# Hybrid Reliable Clustering Algorithm with Heterogeneous Traffic Routing for Wireless Sensor Networks

**DOI:** 10.3390/s25030864

**Published:** 2025-01-31

**Authors:** Sreenu Naik Bhukya, Chandra Sekhara Rao Annavarapu

**Affiliations:** 1Department of Computer Science and Engaging, National Institute of Technology Calicut, NIT Campus P.O., Kozhikode 673 601, Kerala, India; 2Department of Computer Science and Engineering, Indian Institute of Technology (Indian School of Mines) Dhanbad, Dhanbad 826004, Jharkhand, India; acsrao@iitism.ac.in

**Keywords:** wireless sensor networks (WSNs), congestion, priority-based data delivery, trust, K-Harmonic Means (KHM), Enhanced Gravitational Search Algorithm (EGSA)

## Abstract

Wireless sensor networks (WSNs) are vulnerable to several challenges. Congestion control, the utilization of trust to ensure security, and the incorporation of clustering schemes demand much attention. Algorithms designed to deal with congestion control fail to ensure security and address challenges faced due to congestion in the network. To resolve this issue, a Hybrid Trust-based Congestion-aware Cluster Routing (HTCCR) protocol is proposed to effectively detect attacker nodes and reduce congestion via optimal routing through clustering. In the proposed HTCCR protocol, node probability is determined based on the trust factor, queue congestion status, residual energy (RE), and distance from the mobile base station (BS) by using hybrid K-Harmonic Means (KHM) and the Enhanced Gravitational Search Algorithm (EGSA). Sensor nodes select cluster heads (CHs) with better fitness values and transmit data through them. The CH forwards data to a mobile sink once the sink comes into the range of CH. Priority-based data delivery is incorporated to effectively control packet forwarding based on priority level, thus decreasing congestion. It is evident that the propounded HTCCR protocol offers better performance in contrast to the benchmarked TBSEER, CTRF, and TAGA based on the average delay, packet delivery ratio (PDR), throughput, detection ratio, packet loss ratio (PLR), overheads, and energy through simulations. The proposed HTCCR protocol involves 2.5, 2.3, and 1.7 times less delay; an 18.1%, 12.5%, and 5.5% better detection ratio; 2.9, 2.6, and 1.8 times less energy; a 2.2, 1.9, and 1.5 times lower PLR; a 14.5%, 10.5%, and 5.2% better PDR; a 30.7%, 28.5%, and 18.4% better throughput; and 2.27, 1.91, and 1.66 times lower routing overheads in contrast to the TBSEER, CTRF, and TAGA protocols, respectively. The HTCCR protocol involves 4.1% less delay for the ‘$C_{1}$’ and ‘$C_{2}$’ RT packets, and the average throughput of RT is 10.4% better when compared with NRT.

## 1. Introduction

Wireless sensor networks (WSNs) find applications in numerous areas, including disaster management and prevention, agriculture, battlefield surveillance, home security, industrial control, traffic management, healthcare, and many more [1]. WSNs are infrastructureless networks equipped with cost-effective sensor nodes that monitor the surrounding environment. These sensor nodes include sensors, a power supply, microcontrollers, and communicating devices. These units are responsible for data collection, processing, and transmission to the respective sensor nodes through communicating devices. The nodes are arbitrarily distributed and are battery powered, meaning their processing abilities are hugely constrained. Energy constraints, the communication radius, bandwidth problems, computation issues, and the storage capacity of the nodes are significant challenges that need to be addressed. Moreover, these constraints may lead to issues related to scheduling, synchronization, security, network connectivity, coverage, and localization [2]. Sensor nodes with non-rechargeable batteries retain energy for a certain period of time. Ensuring energy efficiency is the main objective, focusing on extending network lifetime by choosing optimal nodes as cluster heads (CHs) so as to support security-aware routing [3]. Several approaches are available to establish secured routing. To facilitate energy saving and the precise management of large-scale networks, clustering methods can be preferably used for achieving secure communication between sensor nodes. However, traditional clustering methods trust every sensor node and treat them as reliable, resulting in selecting malevolent nodes as CHs [4]. Malicious nodes chosen as CHs will influence the processing abilities and security of the network.

Several routing schemes for routing data with improved security are available [5,6,7]. These routing methods are broadly categorized based on the structure of the network as location-, flat-, and hierarchy-based protocols. Location-based routing algorithms are based on sensor node location, while flat-based algorithms offer equal roles to every node depending on the functionality. In the case of hierarchy-based routing algorithms, every node occupies a dissimilar role. Traditional routing approaches forward traffic through unreliable routes, introducing complexities. Congestion detected from the source to the sink in the upstream direction is another challenge that can occur in cases where the speed of transmission and data processing time fall behind the incoming traffic speed. Network congestion may lead to increased latency, packet dropping, buffer overflows, energy wastage, reduced throughput, and decreased quality of service (QoS). The overall network functioning may drop in cases where there is an increase in network congestion. Hence, detecting these congestions and controlling them are basic requirements for WSNs. Though various congestion control protocols are presented, they are not ideal for resource-limited WSNs. So, energy-efficient and dependable congestion detection, control, and transport mechanisms are essential for optimizing QoS needs and network resources. Most of the presented congestion-controlling methods fail to examine the influence of attacks and the role of adversaries in network congestion. Most often, the low-cost nodes are susceptible to breakdown and occasionally become defective. Such nodes are said to be malicious as they operate smartly, giving rise to various security attacks in WSNs that cannot be easily detected. Some security attacks, like Sybil, flooding, and node replication attacks, as discussed in previous studies, have a direct influence on network congestion [8]. They give rise to congestion by creating intermittent network jams, initiating faulty node detection, creating traffic with hoax messages, and re-forwarding the same message multiple times. They cause increased communication overheads, computation time, and energy utilization, thus decreasing network lifetime.

The constraints in WSNs lead to various anomalous attacks, which further affect the regular processing of the network. Such attacks may create physical damage to sensor nodes, resulting in traffic crashes, which involve the dropping of packets, invalid transmissions, and channel jamming through radio intervention. Attacks, which attempt to disturb the routing operations in WSNs [9], can be handled using multi-hop techniques. Though conventional security-based approaches, including encryption and authentication are highly effective in alleviating some attacks, they are not ideal because of their own features [10]. Thus, an effective security-aware routing algorithm to deal with these challenges is required. Trust-based approaches are gaining attention in resolving these challenges to a greater extent. To secure the sensor nodes from malicious attacks and segregate trusted nodes from compromised nodes, trust-based security mechanisms are introduced [11]. These mechanisms aid in detecting trusted nodes, which are highly effective in identifying compromised nodes in the network, as the assessment of trust is associated with the former behavior of malicious nodes and data from neighboring reliable nodes. Due to their logical characteristics, trust-aware routing mechanisms are presented to enhance the security of the network. The selection of optimal intermediary nodes using trust values for reliable routing is a significant deliberation seen in the design of associated scenarios. Furthermore, some trust-based systems with additional factors that include the cost of energy [12], the distance from adjacent nodes to the BS [13], and hop counts [14] are utilized to support secured routing with optimal quality. However, trust-based routing mechanisms face some challenges, as listed below.

Some schemes compute trust based on energy and data but ignore authorization as a reliable standard for using trust values. Hence, the compromised nodes that collaborate with one another or decide to be self-centered are not discarded.Smart compromised nodes are seen to be highly probable in integrating the frequency of malicious attacks with mutable intensity, thus reducing the reliability and efficacy of secure routing protocols. Some trust-based schemes do not consider these threats, though they are found to be capable of circumventing these attacks for some scenarios.Some security schemes implement updating mechanisms and the maintenance of routes to generate fresh routes in case an attack is encountered. However, they fail to assess the possible reasons for the breakdown of routes that have an impact on the efficacy of the update mechanism.

Hence, to overcome shortcomings of traditional cluster-based routing schemes, congestion detection and control mechanisms, and trust-based secure routing schemes, a trust-aware congestion control protocol based on cluster routing is proposed. In this paper, malicious nodes that cause network congestion are effectively identified using an optimized cluster-based routing technique. Further, the hybrid K-Harmonic Means (KHM) combined with the Gravitational Search Algorithm (GSA) is proposed for determining the trust factor, queue congestion status, distance from the BS (sink), and residual energy (RE).

The major contributions of this research paper can be summarized as follows:To provide higher detection accuracy of malicious nodes, we utilize a fitness function that filters the malicious nodes from the WSN. We detect and isolate the malicious nodes from the routing path of the data by utilizing the trust concept. The parameters used in the fitness function are the trust value, link lifetime, energy, and congestion. The computed fitness function detects malicious nodes that cause network congestion and eliminates them from the network.The computed fitness function is applied to the hybrid K-Harmonic Means and Gravitational Search Algorithm. The fitness function in the hybrid algorithm is responsible for the CH selection. The uncompromised nodes other than malicious nodes remain in the network, and the optimal node is elected as the CH. The K-Harmonic Means algorithm is capable of running in the local optima quite easily, thus overcoming the slow convergence speed of the gravitational search algorithm. Hence, these two techniques are merged to function as an effective cluster-based and congestion-free secure routing protocol.We further incorporate a priority-based delivery to the presented secure routing model by monitoring some parameters in the traffic classification model. The parameters that need to be monitored include the humidity, pressure/heat, image/video information, and continuously monitored video information. Humidity and pressure/heat are scalar data. The type of traffic can be identified through these parameters as real-time or non-real-time. Every class of traffic is allocated a dissimilar weight, since every metric is important at different levels. The queuing model carries a threshold value to keep the arriving data packets on the basis of priority queues. This implementation manages network congestion effectively by improving the overall performance of the network.

This paper is structured as follows. Works of different authors are reviewed in Section 2. Section 3 includes the system model and its overview. Section 4 gives the particulars of the algorithm used in the proposed system. Section 5 offers analysis of simulation results. Section 6 gives the conclusion.

## 2. Related Work

Network congestion and various security attacks are some of the significant challenges in resource-limited WSNs. Several conventional algorithms were put forward for energy-efficient routing in WSNs [3,4]. However, these studies fail to examine the real-time challenges that occur because of the network congestion and malicious nodes in WSNs. Congestion-aware, trust-based, secure routing protocols in WSNs are also available in the literature [5,7]. In this section, several clustered routing schemes designed for WSNs are discussed.

Yang et al. [15] focused on ensuring security in clustered industrial WSNs (IWSNs) by presenting a Secure Clustering with Fuzzy Trust evaluation and Outlier detection (SCFTO) protocol. This scheme uses a Markov chain involving two states for modeling the channel quality of the wireless medium. It deals with uncertainty in transmission by estimating trust using an Interval Type-2 (IT2) Fuzzy Logic (FL) Controller. It applies density-based outlier identifications for dynamically acquiring trust thresholds to prevent malevolent nodes from becoming CHs. It employs fuzzy-based CH selection to attain a balance between energy and security, such that a node with an increased amount of residual energy (RE) or reduced assurance on others offers a better probability of becoming the CH. The proposed scheme is capable of dealing with internal malevolent nodes offering better performance based on the number of malevolent clusters, network lifespan, attacks, and throughput. Khan and Singh [16] designed a trust model to examine the reliability of sensor nodes to enhance reliability and data quality and avoid damage as well as adverse effects. A reliable routing approach using a hybrid trust model is proposed to remove selfish nodes from the network. The proposed Trust-Aware Secure Routing Protocol (TASRP) involves several factors like trust, RE, and path length to offer trusted routing paths between reliable nodes involving less energy. This approach aids in selecting reliable nodes for forwarding data, involving less energy owing to its shorter paths. The proposed mechanism offers better trust, throughput, and PDR, involving less energy.

Hu et al. [17] propounded a Trust-Based Secure and Energy-Efficient Routing (TBSEER) protocol to deal with trust and energy in the network. This protocol determines total trust by using dynamic direct, indirect, and energy trusts that may be resilient to selective forwarding, black hole, and hello flood attacks. Furthermore, it employs a penalty scheme along with a volatilization factor for the purpose of quickly identifying malevolent nodes. The nodes compute direct trust, while the sink determines the indirect trust from direct trust sent by nodes, thus reducing the amount of energy consumed as well as congestion due to iterative computations. CHs determine reliable multi-hop paths based on total trust, thus overcoming wormhole attacks. The proposed scheme involves reduced energy, speeds up detection of malevolent nodes, and overcomes common attacks. Chada and Gugulothu [18] focused on supporting reliable data transmission involving reduced energy. Trust-Centric Multi-objective Black Widow Optimization (TC-MBWO) is propounded for choosing secure CHs (SCHs) and trusted paths using MBWO. It determines reliable paths based on cost function factors like RE, trust, distance, as well as node degree. It supports reliable clustering as well as routing, provides resistance to malevolent nodes, and reduces the amount of energy consumed by determining the shortest paths. It offers better results in terms of the quantity of alive and dead nodes, energy consumed, throughput, and network lifespan.

Al-Sadoon and Jedidi [19] proposed the Secure Trust-Aware (ST2A) routing protocol, which offers reliable and trusted routing. The protocol focuses on establishing communication paths based on trust along with an efficient CH selection algorithm during hierarchical routing. It examines the history and state of connections between nodes to assess trust metrics among them. It establishes connections between SNs based on past connections, RE, location, and the state of the node. This trust-aware routing scheme offers improved security and optimizes metrics associated with unique features of the network. The proposed mechanism offers an improved network lifetime when compared with benchmarked schemes. Han et al. [20] proposed the Trust-aware routing protocol using the Adaptive Genetic Algorithm (TAGA) to overcome common routing attacks and also reduce the amount of energy consumed. This protocol determines total trust based on direct trust involving volatilization and dynamic penalty factors and indirect trust using filtering schemes. It offers a threshold function for choosing ideal CHs based on dynamic variations of nodes’ total trust and RE. It employs REs with dynamic crossover and mutation probabilities for finding ideal secure routing for CHs. The proposed scheme reduces the amount of packets dropped by malevolent nodes, thus reducing their impact and offering improved energy efficiency.

Hosseinzadeh et al. [21] propounded the Cluster-based Trusted Routing scheme using Fire Hawk Optimizer (CTRF), which focuses on ensuring restricted energy. It uses a weighted trust mechanism (WTM) based on interaction between sensor nodes. It considers exponential coefficients for trust factors, including the energy state and weighted reception rate and redundancy rate, so that trust of nodes is increased or reduced exponentially depending on friendly or hostile behaviors. It produces a Fire Hawk Optimizer (FHO)-based clustering scheme for choosing CHs from candidate sets that include nodes with RE and trust more than mean RE and trust of all nodes. It uses a cost function based on CH location, energy, distance from the CH to the BS, and the size of the cluster. It determines routing paths between clusters using reliable routing and uses them to transmit data from CHs to BS. It builds routes based on energy, the quality and reliability of routes, and the quantity of hops. It offers improved performance in terms of throughput, energy, packet loss rate (PLR), detection ratio, accuracy, and latency. Osamy et al. [22] proposed the Trust-Aware Clustering Technique based on Improved Rat Swarm Optimizer (TACTIRSO) for an intelligent transportation system (ITS) using WSN. This uses RSO for efficiently selecting trusted CHs. It forms a fitness function based on the RE and trust of nodes. It uses an exponential moving average model to dynamically modify the pre-defined threshold based on the network state. For improved performance, it applies diverse local search approaches along with a trust- and energy-based scheme to initialize the rat population. The proposed mechanism offers better results in terms of energy efficiency, the selection of reliable nodes, the mean network lifetime, and the network stability.

Sharma and Modani [23] proposed a Trust Cluster-based Secure Routing (TCSR) framework by using optimization algorithms. This framework forms clusters by using the Modified Tug of War Optimization (MTWO) algorithm that offers load-balanced clusters to support energy-efficient transmission of data. It selects ideal CHs based on the Received Signal Strength Indicator (RSSI), the rate of congestion and data loss, and throughput. It optimizes parameters using the Butterfly Optimal-Deep Neural Network (BO-DNN) that offers improved security in the selection of CHs. It uses lightweight signcryption for encrypting data that are transmitted between nodes during the transmission of data. The proposed model aids in estimating the trust of every path to help the source in choosing the most reliable one. Nodes enhance reliability as well as security by sustaining the reliability component. Kranthikumar and Leela Velusamy [24] focused on determining an ideal path with a maximum reliability level. The proposed fuzzy and reliable clustering scheme is propounded for enhancing energy efficiency as well as security. Clustering ensures energy efficiency as it employs trust-based FL for identifying malevolent nodes and choosing a trusted path for data transmission. The proposed scheme involves reduced energy and delay, thus offering improved network lifetime, PDR, and security.

Singh et al. [25] focused on ensuring the trusted exchange of data and reliable device communication by offering a trust-based optimization scheme. This scheme manages trust in routing protocols by using Dempster–Shafer Theory (DST), a decision-making algorithm to ensure reliable clustering, and uses Whale Optimization Algorithm (WOA) to support routing. It ensures reliability as well as energy efficiency by confirming trustworthiness in ideal cluster and path selection. It offers an improved network lifespan, throughput, and energy efficiency. Bai et al. [26] proposed the Trust-based and energy-aware Secure Routing Protocol (TSRP), a multi-dimensional trust assessment framework. The proposed system assesses trust that is indispensable for guaranteeing the security of nodes. Rather than trusting exclusively in encryption, it allocates trust to relay nodes and obtains the reputation from neighboring nodes. It efficiently finds trusted relay nodes when reducing the effect on resources. The proposed authentication scheme involves an advanced critical production scheme and guarantees resource efficacy by calculating multi-dimensional trust. It aids in creating an equilibrium between enhancement in resource usage and the reinforcement of security. It ensures energy efficiency, reduces energy fluctuations, and offers increased speed. Considering all the above existing studies in the literature, it is evidently clear that an efficient design of a congestion-aware secure routing method using hybrid KHM and GSA will overcome all the significant challenges faced by these studies. A clustering scheme is included in the proposed secure routing protocol. Further, priority-based routing is incorporated to effectively avoid network congestions and promote the free flow of network traffic.

## 3. System Model

A WSN is represented as a graph G(N,C), where $N=\{n_{1},\;n_{2},\;n_{3},\;\cdots ,\;n_{t}\}$ signifies a collection of nodes and ‘*t*’ denotes a set of connections between nodes $n_{i}$ and $n_{j}$. Consider that nodes lie within the coverage area (A) of a WSN. Every connection will be based on the delay that occurs while transmitting data and also based on the distance between ‘$n_{i}$’ and ‘$n_{j}$’. Assume ‘S’ as the transmitting node that forwards data to the intended destination (D) through intermediary nodes. Our congestion-free, trust-aware, secure routing protocol ensures data security based on trust and reduces network congestion through priority-based data delivery. In the proposed scheme, malicious nodes are circumvented so as to ensure trust. They are not involved in communication, thus reducing the amount of traffic. Further traffic load is taken into consideration when determining the cost function. Furthermore, traffic is prioritized to avoid delay. All these factors contribute to security and reduction in congestion in the network. Figure 1 shows the communication of data in a typical WSN.

Parameters such as trust, waiting time, energy, and distance are taken into consideration. These parameters, referred to as weights, are used in the determination of FF, which aids in the selection of optimal routes based on application demands. The classification of traffic enables the data traffic to be prioritized depending on the significance of application needs. FF is based on hybrid KHM and the GSA algorithm, as discussed in Section 4, and is used for optimal route selection based on node quality and data traffic. The following sections describe the method for determining fitness parameters. Further, the notations used are listed in Table 1.

### 3.1. Trust Model

In cases where the distance of the adjacent node is less than the transmission range, the trust of the node needs to be determined. RSSI is determined based on distance.(1)$${\mathrm{Dist}}_{s,i}=\mathrm{RSSI}\left(N,G_{i}\right)$$
where *N* refers to the input node, ${\mathrm{Dist}}_{s,i}$ refers to the signal strength between the source node *S* and the current node *i*, and $G_{i}$ refers to the $i^{th}$ node in the graph.

Trust can be calculated using direct as well as indirect trust values, as shown below:(2)$${\mathrm{Trust}}_{xy}\left(i\right)=\omega {\mathrm{Trust}}_{xy}^{DT}\left(i\right)+\gamma {\mathrm{Trust}}_{xy}^{IDT}\left(i\right)$$
where ${\mathrm{Trust}}_{xy}^{DT}\left(i\right)$ refers to the degree of direct trust that exists within a node at time *i*, ${\mathrm{Trust}}_{xy}^{IDT}\left(i\right)$ refers to the degree of indirect trust that exists within a neighboring node at time *i*, and ω and γ are the fitness factors that range between 0 and 1, such that ω+γ=1.

The degree of DT can be estimated based on packets sent and received at a node.(3)$${\mathrm{Trust}}_{xy}^{DT}\left(i\right)=\frac{T_{xy}^{RX}\left(i\right)}{T_{xy}^{SD}\left(i\right)}$$
where $T_{xy}^{RX}\left(i\right)$ denotes the number of packets received by the node *x* at time *i* and $T_{xy}^{SD}\left(i\right)$ denotes the total number of packets sent by the node *y* at time *i*. The degree of IDT is based on the transmission of the packet to neighboring nodes, as shown below.(4)$${\mathrm{Trust}}_{xy}^{IDT}\left(i\right)=\frac{1}{N_{d}}\sum_{d=1}^{N_{d}}{\mathrm{Trust}}_{dy}^{DT}\left(i\right)$$
where ${\mathrm{Trust}}_{dy}^{DT}\left(i\right)$ refers to the trust degree that exists in the neighbors at time *i* and $N_{d}$ refers to the number of neighboring nodes.

This factor improves the degree of trust by ignoring anomalous nodes and choosing nodes with increased trust for forwarding data packets with complete trust. In every iteration, the degree of trust is determined by using the moving average model as stated below:(5)$${\mathrm{Trust}}_{xy}^{*}\left(i+1\right)=\tau \cdot {\mathrm{Trust}}_{xy}\left(i\right)+\left(1-\tau \right)\cdot {\mathrm{Trust}}_{xy}\left(i+1\right)$$
where τ is the fitness factor utilized to stabilize the computations in the recent and the past iterations and is specified in the range 0<τ<1, while ${\mathrm{Trust}}_{xy}^{*}\left(i+1\right)$ is the degree of trust computed at time i+1. Trust values are recorded as a table that contains information related to the trust of every neighbor involved in data forwarding. The node detection level ($NDL_{y}$) is determined as follows:(6)$${\mathrm{NDL}}_{y}=\frac{1}{{\mathrm{Trust}}_{xy}^{*}\left(i\right)}$$

$NDL_{y}$ can be either normal or the attacker, as shown in the following conditions.

If $0\leq NDL_{y}\leq 0.7$, the node will be considered a normal node.If $0.8\leq NDL_{y}\leq 1$, the node will be considered an attacker node.

### 3.2. Energy Model

The energy model considers several attributes such as transmission energy, reception energy, sleep time energy (energy spent when the node sleeps), and listening energy. The quantity of energy consumed by the sensor node $N_{CE}$ is determined as follows:(7)$$\begin{array}{c} \sum_{k=0}^{l}N_{CE}^{k}=\sum_{k=0}^{l}N_{TXE}^{k}\times t_{RX}^{k}+\sum_{k=0}^{l}N_{RXE}^{k}\times t_{TX}^{k}+\sum_{k=0}^{l}N_{LXE}^{k}\times t_{LX}^{k}+\sum_{k=0}^{l}N_{SXE}^{k}\times t_{SX}^{k} \end{array}$$

$N_{RE}^{k_{t_{0}}}$ and $N_{RE}^{k_{t_{1}}}$ represent the available energy of the sensor node $N_{k}$ and $Energy\_res(N_{i},t_{1})$ represent the available energy of the node $N_{i}$ at time ‘$t_{0}$’ and ‘$t_{1}$,’ respectively. The quantity of energy consumed during transmission as well as the reception of data are denoted as ‘$N_{TXE}$’ and ‘$N_{RXE}$,’ respectively. In cases where there is no data transmission, then the node will be moved to a sleep or listening state, denoted as ‘$N_{LXE}$’ and ‘$N_{SXE}$,’ respectively.

The total residual energy can be calculated based on the initial energy ‘$N_{lE}$’ and the amount of energy consumed ‘$N_{CE}$’ as follows:(8)$$\sum_{k=0}^{l}N_{RE}^{k}=\sum_{k=0}^{l}N_{IE}^{k}-\sum_{k=0}^{l}N_{CE}^{k}$$

### 3.3. Traffic Load Calculation

To reduce energy overheads, the traffic load for nodes with a trust value that is higher than the value of the trust threshold is identified. Further, to detect network congestion, ‘$CongTh_{Min}$’ and ‘$CongTh_{Max}$’ thresholds that fall within the range of buffer length in the queue are set. Depending on the fixed threshold values, network congestion can be classified into three types, as follows:Less: If the queue length falls below the threshold $congTh_{\min}$, the network is considered as having congestion.Medium: If the queue length is between $congTh_{\min}$ and $congTh_{\max}$, the network is considered as having medium congestion.High: If the queue length is higher than the threshold $congTh_{\max}$, then the congestion level is considered as being high.

Let $T_{s}^{k}$ represent the buffer length of the ‘ $k^{th}$’ node, represented as(9)$$CI_{k}=f\left(T_{s}^{k}\right),CI_{k^{\prime }}=1-CI_{k}$$
where $T_{s}^{k}=CI_{k}$.

If $T_{s}^{k}=CongTh_{Min}$, the quality of congestion will be lower, where *C* refers to a lower quantity.

If $T_{s}^{k}=CongTh_{Min}\leq T_{s}^{k}\leq CongTh_{Max}$,$$CI_{k}=1-C\cdot \frac{T_{s}^{k}-CongTh_{Min}}{CongTh_{Max}-CongTh_{Min}}+C$$

If $T_{s}\left(k\right)>congTh_{\max}$, the congestion is high, and it becomes 1.

### 3.4. Distance Calculation

The distances between the sender, the potential next node, and the sink are computed. Let $dist_{1}$ represent the distance between the recent source and the probable next nodes and let $dist_{2}$ denote the distance between the next node and the sink. The matching distance from the sender $dist_{1}^{M}$ and the matching distance from sink $dist_{2}^{M}$ are considered.(10)$$dist_{1}^{M}=1-dist_{1},\;dist_{2}^{M}=1-dist_{2}$$

Hence, the distance of the potential next node associated with matching distances $dist_{1}^{M}$ and $dist_{2}^{M}$ can be determined as follows:(11)$$T_{dist}=\frac{\alpha \cdot dist_{1}^{M}+\beta \cdot dist_{2}^{M}}{\alpha +\beta }$$
where α and β are the adjustment parameters for $dist_{1}^{M}$ and $dist_{2}^{M}$.

### 3.5. Waiting Time of the Nodes

The waiting time at each node in the cluster is computed to find out whether a node in the cluster can be selected as CH or not.(12)$$WT=WT_{max}\times \omega \left(1-\frac{N_{RE}}{N_{lE}}\right)\times \sigma \left(|\mathrm{Avg}\left(S_{i}^{k}\right)-S_{n}|\right)$$
where $WT_{max}$ represents the maximum pre-defined time to wait, $\mathrm{Avg}(S_{i}^{k})$ denotes the average speed of adjacent nodes within the range of transmission, and $S_{n}$ signifies the speed of the node.

## 4. Hybrid Trust-Based Congestion-Aware Cluster Routing (HTCCR)

In this section, the proposed HTCCR protocol using hybrid KHM and the EGSA is detailed below. The complete flow of the proposed work is shown in Figure 2.

The proposed HTCCR protocol considers various QoS parameters like energy, trust, distance, and traffic load for fitness selection as a multiple-objective system. Once the fitness parameters are computed, it discovers k-probable paths between the source and destination. The primary objective of the propounded algorithm is to choose the best route depending on several objectives that are considered to support priority-based data delivery along with secure routing. Priority-based data delivery is explained in detail in the ensuing sections. Further, to ensure security, data are encoded using XOR encoding before they are transmitted to the destination, and XOR decoding is performed at the destination to obtain data.

### 4.1. CH Selection Phase

During CH selection, a cost function based on energy, distance, trust, and load level is employed. These values are considered as fitness values for computing the cost function and they are given as the input to the hybrid KHM and GSA. A node with an ‘NDL’ greater than 0.8 is considered to be an attacker. In such a case, the attacker will not be allowed to take part in the process of CH selection.(13)$$\mathrm{Overall}\;\mathrm{fitness}=T_{\mathrm{dist}}\times \delta +CI_{k^{\prime }}\times \varepsilon +WT\times \vartheta +\left(\left(\mathrm{Trust}\times \gamma \right)+N_{\mathrm{RE}}\times \tau \right)$$
where δ, ε, ψ, γ, ϑ, and τ are the adjustment parameters that lie between 0 to 1.

The overall fitness of the energy level recommends that nodes with increased distances include a higher value of overall fitness. In this scenario, those nodes with a higher value of ‘overall fitness’ have the lowest possibility of becoming CHs. Conversely, the nodes that have a large number of neighbors and the nodes with increased energy will have a lower value of ‘overall fitness.’ The nodes with a lower value of ‘overall fitness’ will have a greater possibility of becoming CHs. Further, they aggregate the data to reduce redundancy. The aggregation of data conserves energy within the network since unnecessary data are not transmitted to the BS (sink). Once data are collected and a CH is chosen, every node with a specific energy level sets its ‘overall fitness’ to a particular value. A node with a lower ‘overall fitness’ value involves less energy consumption and has the possibility of becoming a CH. The chosen CH forwards data packets to other nodes within its radius.

#### 4.1.1. Hybrid K-IGS Algorithm

The hybrid K-IGS algorithm considers QoS parameters (fitness) like the packet delivery ratio (PDR), energy, distance, and traffic load as multiple objectives for selecting the best route with improved security. The hybrid K-IGS algorithm is a widely used optimization clustering method that is extensively applied in various fields.

#### 4.1.2. KHM Clustering

This clustering method is widely used because it is simple and can be directly implemented involving a lower number of iterations. The K-means (KM) clustering method strives to identify the cluster centers (${\mathrm{cen}}_{1}$, ${\mathrm{cen}}_{2}$, ${\mathrm{cen}}_{3}$, ..., ${\mathrm{cen}}_{n}$), in which the total squared distances of every data point ${\mathrm{dp}}_{i}$ to the cluster center ${\mathrm{cen}}_{j}$, which is too close, are curtailed. The performance of the KM method depends on cluster center initialization. The reason behind this problem is that the KHM method utilizes partitioning plans that lead to the robust integration of the closest data center and data points, forbidding data centers from moving away from the localized data density. The KM method utilizes the shortest distance from data points to data centers, and KHM clustering substitutes this with the HM of distances from data points to every data center. The KHM method provides the best and lowest score for data points when they are very near to a data center. This is considered a significant feature of the KHM method, which similar to like that of the minimum function employed in the KM method. The KHM clustering method is designed as shown below.

Assume $CC=({\mathrm{cc}}_{1},\;{\mathrm{cc}}_{2},\;\cdots ,\;{\mathrm{cc}}_{n})$ as the cluster center set, and $D=(d_{1},\;d_{2},\;\cdots ,\;d_{m})$ as the data considered for clustering, $\mathrm{mem}({\mathrm{cc}}_{j}|d_{i})$ as a membership function that defines the number of data points that are part of the data center ${\mathrm{cc}}_{j}$, and $\mathrm{wei}(d_{i})$ as the weight function that defines the number of data points $d_{i}$ present in recalculating the data center parameters in the upcoming iteration.

**Algorithm:** KHM Clustering Method

Initialize m by arbitrarily selecting data centers CC.Compute the value of the objective function depending on(14)$$KHM\left(D,CC\right)=\sum_{i=1}^{m}\frac{n}{\sum_{j=1}^{n}\frac{1}{∥d_{i}-cc_{j}{∥}^{r}}}$$
where r≥2 is an input parameter.For every data point $d_{i}$, calculate the membership function $\mathrm{mem}(cc_{j}|d_{i})$ for each data center $cc_{j}$ depending on(15)$$\mathrm{mem}\left(cc_{j}|d_{i}\right)=-\frac{∥d_{i}-cc_{j}{∥}^{-(r+2)}}{\sum_{j=1}^{n}{∥d_{i}-cc_{j}∥}^{-(r+2)}}$$For every data point $d_{i}$, calculate the weight function $\mathrm{wei}(d_{i})$ depending on(16)$$\mathrm{wei}\left(d_{i}\right)=\frac{\sum_{j=1}^{n}{∥d_{i}-cc_{j}∥}^{-(r+2)}}{{\left(\sum_{j=1}^{n}{∥d_{i}-cc_{j}∥}^{-(r+2)}\right)}^{2}}$$For every data center $cc_{j}$, recalculate their location from all the data points $d_{i}$ depending on their weight and membership values:(17)$$cc_{j}=\frac{\sum_{i=1}^{m}\mathrm{mem}\left(cc_{j}|d_{i}\right)\;\mathrm{wei}\left(d_{i}\right)\;d_{i}}{\sum_{i=1}^{m}\mathrm{mem}\left(cc_{j}|d_{i}\right)\;\mathrm{wei}\left(d_{i}\right)}$$Repeat Step 2 to Step 5 until KHM(D,CC) does not experience any significant variation or for a fixed number of iterations.Allocate a data point ‘$d_{i}$’ to cluster ‘*j*’ with the largest membership value ‘$\mathrm{mem}(cc_{j}|d_{i})$. KHM has been found to be insensitive to the initialization of data centers when it is prone to local optima convergence [27].

#### 4.1.3. Enhanced GSA (EGSA)

In the GSA [28], the objects moving out of bounds will be allocated a boundary value. This algorithm is prone to the local optima problem in cases where there are more objects present on the boundary. Specifically, if few local optimal values are present on the boundary, the algorithm stays in the local optima for some time and will be deprived of its population diversity. Hence, Yin et al. [29] presented an improved gravitational search algorithm to resolve this problem in the original GSA. Further, they also presented a hybrid algorithm of KHM and IGSA as a clustering algorithm. We utilize this hybrid algorithm in our proposed study.


**Hybrid clustering algorithm of KHM and EGSA:**
Fix the initial parameters that contain the highest iterative count IteCnt, the size of the population, pop_size, and $Ge_{0}$.Now, initialize the population size pop_size.Fix the count for iterations $Gener_{1}=0$.Similarly, fix the count for iterations $Gener_{2}=Gener_{3}=0$.Use the IGSA technique as in Yin et al. [29].(a)Utilize the IGSA operator for updating the pop_size objects.(b)$Gener_{2}=Gener_{2}+1$. If $Gener_{2}<8$, go to step 5(a).Utilize the KHM technique (Equations (14)–(17)). For every object *i*,(a)Consider the position of object *i* as the cluster center for the KHM technique.(b)Recompute every cluster center by employing the KHM technique.(c)$Gener_{3}=Gener_{3}+1$. If $Gener_{3}<4$, go to step 6(b).$Gener_{1}=Gener_{1}+1$. If $Gener_{1}<IteCnt$, go to step 4.Allocate the data point $d_{i}$ to the cluster *j* with the highest membership value $mem(cc_{j}|d_{i})$.


#### 4.1.4. Cluster Maintenance Phase

The cluster maintenance phase instantly begins once the first cluster is formed. This phase begins when any of the listed problems are encountered. The proposed algorithm provides a solution to every issue addressed below.

**If a CH fails due to a discharged battery:** If this issue occurs, the following steps can be taken.(a)Nodes can combine with other CHs through CH and member selection.(b)Selecting a fresh CH from the sustained nodes and by attracting other nodes to the latest cluster.**If the members fail because of battery discharge:** If this occurs, the CH ignores the dead nodes. Further, it transmits messages to the members and examines their efficacy and modified characteristics. In cases where the node fails to respond with any message, the node is considered to be dead, or it may be outside the range of the cluster. Hence, the node is removed from the member list.**CHs may interfere:** In this case, the fitness values of two interfering CHs are compared, and the one with the highest value is selected as the CH. Nodes belonging to the CH with the lowest value join the new cluster. The non-member nodes of the new CH are chosen depending on the values obtained, and the other nodes will combine with the new CH.


**Data forwarding phase**


When forwarding data, the source forwards its data to the CH within its range. The CH obtains data and forwards them to the nearest CH or to the sink in cases where the sink falls within the CH range. The priority-based scheduling of data packets and network coding are introduced to improve data delivery. These phases are detailed below.

## 5. Priority-Based Data Delivery

This phase includes application, routing, queuing, and neighboring node modules. The application module categorizes the incoming data packets into control and data packets, respectively. Data traffic is determined based on QoS demands. The packets can be classified as (i) reliable, (ii) normal, (iii) delayed, and (iv) critical. Control packets are transmitted to neighboring nodes such that every node locally updates the neighbors with the routing decision. If a packet of data is broadcast, this application module enables the following routing module to establish a route based on packet type. The queuing module is responsible for allocating priority to every packet before sending the packet for channel access. The application module utilizes a traffic categorization module as presented in Figure 3. The neighboring node module observes the forwarded packet to share details of node quality and node state among its neighbors. The routing module is responsible for the selection of optimal routes based on the attributes of neighbors that represent node quality. The selection of nodes and CH will be based on the clustering technique and FF. The queuing module is responsible for categorizing packet priority into two types: (i) RT and (ii) NRT queues. Delay-sensitive packets are assigned high priority and are put into the RT queue. The remaining classes of data packets are put into the NRT buffer. The attempt to categorize incoming packets into real and NRT buffers produces reduced latency for data packets within a pre-defined time. Further, the queuing model uses a first-in-first-out (FIFO) procedure for assigning priority to packets. This reduces delays in the data traffic as in Figure 4.

Table 2 shows the type of data traffic in WSNs. Data traffic is classified into four types, namely ‘$C_{1}$’, ‘$C_{2}$’, ‘$C_{3}$’, and ‘$C_{4}$’, where ‘$C_{1}$’ represents high RT traffic that demands less delay and high reliability as it is associated with critical tasks, ‘$C_{2}$’ represents scalar RT packet traffic that requires packet delivery within a pre-defined time for the packet’s arrival at the sink, ‘$C_{3}$’ represents scalar NRT packet traffic that demands delivery without any pre-defined time but permits an acceptable packet loss for the periodic monitoring of applications, and ‘$C_{4}$’ represents best effort (BE) traffic without any particular requirements.

Table 3 shows the weights assigned to traffic due to dissimilar levels of metric significance. The overall weight of fitness factors is ‘1’.

Priority-based data delivery allocates priority to every packet that is forwarded in the communicating medium depending on the traffic type. A multi-queuing priority policy is established to enable the nodes to construct several queues of priority for various classes of traffic by assigning a dissimilar level of importance to each class in a service policy. Every queuing model contains a particular value of threshold for the incoming packets based on the priority queues that handle network congestion effectively and also enhance the overall network performance. The data types involve several levels of importance and quality and therefore include diverse priorities in routing data packets. Hence, to fulfill several QoS parameters based on diverse classes of data, every class is assigned a level of priority, and every packet type should be handled in a different way based on the allocated priority.

Further, traffic categorization can be used as a basis for making routing decisions. The overall delay involved in data packets includes delays associated with transmission, queuing, propagation, and processing. Transmission delay varies with the size of data packets and the data rate of nodes computed from the time of transmission of the opening bit of data to that of the last bit. Queuing delay is the difference in time between the data packet entering the queue and the time of transmission. In cases where the data transmission from a communicating link is higher, the waiting time of the packet will be extended. The propagation delay is usually ignored as it is too short. It relies on the physical features of the communicating medium. The processing delay is the variance in time between when the packet is obtained in the receiving queue and the time the packet is moved to the transmit queue. In cases where there is an increase in network congestion, the overall delay increases, thus influencing the performance of routing to a greater extent.

To overcome these issues, a queuing model that categorizes packets based on priority, either as real- or non-real-time, is included in the proposed system. The responsibility of the traffic classification model is to determine the class of the arriving data packets and forward them to the respective queue. The packets arriving in ‘$C_{1}$’ and ‘$C_{2}$’ with a pre-defined time period are given high priority and put in the RT queue, and those in ‘$C_{3}$’ and ‘$C_{4}$’ are moved to the NRT buffer. To control the data packets so that they are in the right queue based on queuing policy priority and data type, a packet scheduler is designed as shown in Figure 5.

It is assumed that the size of every queue is assigned a threshold value, Thr. A queue with priority ‘a’ for a node ‘k’ includes data packets with lower priority ${\left(NQ_{L}\right)}_{k}$, those with the same priority ${\left(NQ_{a}\right)}_{k}$, and those that are critical ${\left(NQ_{c}\right)}_{k}$. If a packet *P* arrives at the *m*-th queue $Q_{m}$ at node *k*, the anticipated wait time WT computed [11] is decided based on the following: (i) the residual time for a packet that is served currently for delivery ($rt{\left[cur\right]}_{k}$), (ii) the servicing time for arriving packets with high priority or the same priority ${\left(NQ_{a}\right)}_{k}+{\left(NQ_{c}\right)}_{k}$, and (iii) the delivery-based servicing time for the arriving packets of high priority in the course of the packet’s wait time ($wt[NQ_{b}^{k}$]). The number of incoming data packets is less than the threshold (Thr). Hence, the overall anticipated wait time for a packet (P) is given by(18)$${\left[WT\right]}^{k}=rt{\left[\mathrm{cur}\right]}^{k}+\sum_{a,c,b=0}^{Thr}\left(\left(NQ_{a}^{k}\right),\left(NQ_{c}^{k}\right),\left(NQ_{b}^{k}\right)\right)$$

According to Equation (18), the traffic classifier decides on the transmission of data packets as follows: ‘$C_{1}$’ is allotted the highest priority, followed by ‘$C_{2}$’, ‘$C_{3}$’, and ‘$C_{4}$’, which is given the lowest priority. As shown in Figure 6, the queue model involves a particular queue size for every type of packet, thus effectively managing the throughput of every data class.

So as to realize RT data transmission within a particular time period, the proposed scheme fixes a bandwidth ratio (br) that denotes the amount of bandwidth to be dedicated to ‘$C_{1}$’ and ‘$C_{2}$.’ The value of ‘br’ is employed to compute the service rate for ‘$C_{1}$,’ where ‘(1-br)’ denotes the transmitting rate of ‘$C_{2}$.’

In order to realize the real-time data transmission within a particular time period, the proposed scheme fixes a bandwidth ratio br, which denotes the amount of bandwidth to be devoted to the parameters C1 and C2. This value of br is employed to compute the service rate for C1, where (1-br) denotes the transmitting rate of C2.


**Queuing and Average Delay Computation**


To determine the optimal path based on QoS parameters that satisfy the needs for every traffic type, computation of the queuing buffer along with the average delay is essential. The total load on a node ‘a’ is determined as shown below.(19)$${\left(T_{s}\right)}^{m}=PI_{m}\cdot SN_{m}+\sum_{m}^{l}\left(l|m\right)\cdot \lambda_{n}$$(20)$${\left(\lambda_{a}\right)}^{m}=\frac{D_{l}}{D_{r}}+{\mathrm{tim}}_{BOT}$$(21)$${\mathrm{tim}}_{BOT}=r\left({\mathrm{tim}}_{CCA}\right)+\sum_{d=0}^{r-1}{\mathrm{tim}}_{BOA}\left(\mathrm{dMinBE}+d,\mathrm{dMaxBE}\right)$$
where $PI_{m}$ represents the data packet interval, $SN_{m}$ represents the sourcing node, and $\left(l|m\right)\lambda_{n}$ represents the service rate for the neighboring nodes for transmitting data packets, $D_{l}$ represents the data length, $D_{r}$ represents the data transmission rate, ${\mathrm{tim}}_{BOT}$ represents the time for back-off, which relies upon the channel load, ${\mathrm{tim}}_{BOA}$ represents the number of back-off attempts, $r({\mathrm{tim}}_{CCA})$ is the amount of time with clear access to the channel, *d* represents the maximum number of attempts for back-off before decline, the value of dMaxBE=5, and the value of dMinBE=3.

According to Equations (19)–(21), the data load in a node increases when a node receives data from diverse neighboring nodes. Such an increase in data load results in network congestion, while the service rate of a node that forwards data packets is lower when compared with the interval of the arriving packets. Congestion leads to packet loss due to failure in accessing the channel or the capacity of the buffer overflowing beyond the pre-determined threshold value. The overall queuing delay in node ‘a’ is calculated as shown below.(22)(oqueue)m=(Load)m(λa)m=PIm·SNm+∑m=0llm(λn)(λa)m

The queue length for a node is determined as shown in Equation (18). In cases where the packet quantity increases beyond the queue length and its value exceeds the threshold value, the packets will be dropped as follows:(23)pktdrop=PIm·SNm+∑m=0llm(λn)(λa)m>Thrh
where ${\mathrm{Thr}}_{h}$ is the threshold value that keeps the data packets in the queue and ${\left(\lambda_{a}\right)}^{m}$ refers to the RT data transmission rate of a node. The presented protocol allocates a high service rate to the RT traffic categories ${\left(\lambda_{a}\right)}^{m}$ when compared with that of the non-real-time classes ${\left(1-\lambda_{a}\right)}^{m}$.

Considering that the RT service rate ${\left(\lambda_{a}\right)}^{m}$ is equal to $n{\left(1-\lambda \right)}^{m}$, the proposed protocol drops the packet dropping ratio for RT packets as shown in Equation (24).(24)droprate=PIm·SNm+∑m=0llm(λn)(λa)m

To ensure the reliable delivery of RT data packets to the sink from the source, the average queuing delay for a specific path ‘m’ can be computed as follows:(25)(avgdelay)m=∑m∈path(hop)·PIm·SNm+∑m=0llm(λn)(λa)m(26)PImλ=∑m∈path(hop)·SNm+∑m=0llm(n)am

According to Equations (22)–(26), the hop count, or the distance between the node and sink, influences the average delay. As long as the node is in proximity to the sink, it involves less hopping delay and will be a highly beneficial node for delivering RT data packets.

## 6. Data Transmission Using XOR Encoding and Decoding

The data are encoded using XOR encoding before they are transmitted to the destination and the destination performs XOR decoding to retrieve the data.

### 6.1. Network Coding

In this phase, the packet is encoded at the source and forwarded to the CH that is within its range, and the sink decodes the packet. At the source, XOR encoding is employed for encoding data packets, and XOR decoding is performed at the sink for decoding data packets. The process of XOR encoding and decoding is explained below.

Network coding refers to the process of linear encoding and decoding in which the intermediary nodes perform encoding of data packets obtained from neighboring nodes in the underlying network.

### 6.2. Encoding Process

When the hub shares encoded packet bundles with another hub, it collects an array of coefficients, namely $\mathrm{encv}=({\mathrm{encv}}_{1},\;{\mathrm{encv}}_{2},\;\ldots ,\;{\mathrm{encv}}_{n})$, termed as the encoding vector, which is derived from $\mathrm{ENV}(2^{s})$. While a hub collects the array of the *k* bundles, $E_{m}$ (m=1, 2, 3, 4, …, k) is encoded straight away into a structure of a solitary yield packet bundle, which is derived as(27)$$\mathrm{EPB}=\sum_{m=1}^{k}{\mathrm{encv}}_{m}\cdot E_{m},\;\mathrm{where}\;{\mathrm{encv}}_{m}\in \mathrm{ENV}\left(2^{s}\right)$$

The currently encoded packet bundle is forwarded with ‘k’ coefficients in the model. The receiving hub utilizes a homogeneous encoding vector so as to decode the packet bundle.

### 6.3. Decoding Process

On receiving the encoded packet bundles, the receiving hub proceeds to comprehend an array of undeviating situations to acquire the initial packet bundles from the information in encoded form. If an encoded vector ‘v’ is received by the recipient node, the vector is fixed as $\left({\mathrm{encv}}_{1},{\mathrm{EPB}}^{1}\right),\cdots ,\left({\mathrm{encv}}_{p},{\mathrm{EPB}}^{p}\right)$ by the mid-point. The receiving midpoint resolves the undeviating conditions with ‘p’ conditions and ‘k’ queries, aiming to extricate the prevailing condition.(28)$${\mathrm{EPB}}^{j}=\sum_{m=1}^{k}{\mathrm{encv}}_{m}^{j}\times E_{m,j}\;j=1,\;2,\;\cdots ,\;p$$

In any event with *p* un-deviated self-governing encoded data packet bundles, the primary objective is to extricate the packet bundles. The chief ambiguous $G_{m}$ retains the initial data bundles, which are about to be forwarded in the system. *k* distinct bundles are deduced by enlightening the straightforward condition explained in Equation (28) to fetch *k* direct free packet bundles. The XOR encoding scheme is a well-organized coding method and is also a distinctive example of direct system coding followed in this study. The packet bundles that are encoded and used in the system contain a location as the constituent in ENV(2)={0,1}, and the bitwise XOR encoding in ENV(2) is employed in this study. The operation of the packet data processing, encoding, and decoding algorithms is explained below.

Algorithm: packet data processing (Pkt_m) at a node within the network code layer.

Pre-requisites: while the PacketData transmission/reception begins, the received packet PacketData will be included in the ReceiveDataQueue() To be monitored: check whether the encoded PacketData are forwarded or eliminated.

Step 1: collect the PacketData Pkt_m from the ReceiveDataQueue().

Step 2: if the PacketData Pkt_m ∈ EncNodeSet(), proceed.

Step 3: if native (Pkt_m), then.

Step 4: EP=XOREncode().

Step 5: The k node forwards the encoded PacketData EP to the base station (sink node).

Step 6: Enter the processed PacketData Pkt_m to the TxdPacketDataSet();.

Else

Step 7: Eliminate (Pkt_m).

Endif

Else

Step 8: the node k operates as a forwarding node and forwards the PacketData Pkt_m to the sink node.

Endif

Endif

Step 9: if(ReceiveDataQueue()=blank)

Step 10: go to the Step 1.

Else

Step 11: exit.

endif

Algorithm: encoding algorithm XOREncode()

Pre-requisites: ReceiveDataQueue() and the SensedQueue() has to be maintained in the encoder node. To be monitored: encoded PacketData EP generation.

Step 1: If the SensedQueue() is not blank, then proceed.

Step 2: Collect a PacketData Pkt_m from the top of the ReceiveDataQueue().

Step 3: Collect a PacketData Pkt_j from the top of the SensedQueue().

Step 4: EP=Pkt_m⊕Pkt_j.

Else

Step 5: Collect the next data packet Pkt_m+1 from the ReceiveDataQueue().

Step 6: EP=Pkt_m⊕Pkt_m+1.

Endif

Step 7: return EP.

## 7. Results and Discussion

The proposed protocol is simulated using Network Simulator-NS3. The simulation parameters are listed in Table 4. The performance and efficiency of the propounded HTCCR protocol are analyzed and compared with those of the proposed TBSEER (Hu et al. (2021) [17]), CTRF (Hosseinzadeh et al. (2023) [21]), and TAGA (Han et al. (2022) [20]) algorithms.

The following parameters are analysed: PDR, average delay, overhead, average energy consumption, packet loss ratio (PLR), end-to-end delay, average throughput, and detection ratio.

Figure 7 shows the delay involved in forwarding the data packet from the source to the sink. It is obvious that the propounded HTCCR protocol involves the least delay when compared with the benchmarked TBSEER, CTRF, and TAGA protocols. The proposed HTCCR protocol involves 2.5, 2.3, and 1.7 times less delay compared with the TBSEER, CTRF, and TAGA protocols, respectively.

Figure 8 depicts the detection ratio of the proposed HTCCR protocol in a network scenario open to anomalous attackers. It is assumed that the number of attackers snooping data packets is nearly equal to the number of attackers furnishing forged information. The existing TBSEER, CTRF, and TAGA schemes are capable of dealing with partial attacks, and hence maintaining security is tedious. The proposed protocol is efficient in identifying malicious nodes in the network. It offers an 18.1%, 12.5%, and 5.5% better detection ratio compared with the standard protocols investigated in this study. Figure 9 shows the amount of energy consumed during node failures. It is seen that the HTCCR protocol outperforms the standard protocols considered in this study. The proposed protocol involves 2.9, 2.6, and 1.8 times less energy when compared with the benchmarked schemes.

With an increase in the number of malicious nodes, the packet loss ratio (PLR) increases (Figure 10). The proposed HTCCR protocol is found to offer standard performance even when malevolent nodes are observed in the network. It is obvious that the proposed protocol involves a 2.2, 1.9, and 1.5 times lower PLR compared with the standard TBSEER, CTRF, and TAGA protocols, respectively. The PDR decreases with an increase in the number of malevolent nodes. Figure 11 shows the PDR of the proposed and standard schemes considered in this study. It is evident that the HTCCR protocol offers a 14.5%, 10.5%, and 5.2% better PDR in comparison to the TBSEER, CTRF, and TAGA protocols, respectively.

Throughput also shows a decrease with an increase in the percentage of malicious nodes. As seen in Figure 12, the proposed protocol outperforms the benchmarked TBSEER, CTRF, and TAGA protocols by 30.7%, 28.5%, and 18.4%, respectively. Figure 13 shows the routing overheads involved in the proposed and existing schemes. The overheads involved increased with the number of nodes. Malicious nodes also contribute to increased overheads. It is seen that the standard TBSEER, CTRF, and TAGA protocols involve 2.27, 1.91, and 1.66 times higher routing overheads compared with the proposed HTCCR protocol, respectively.

Figure 14 and Figure 15 show the impact of traffic rate on the end-to-end delay and throughput of RT and NRT packets. The average packet delay is the average time necessary for forwarding a packet to the sink from the source. As seen in Figure 14, the delay is directly proportional to the traffic rate, as increasing the PDR produces increased queuing delay. RT packets involve less delay in comparison to NRT packets for varying levels of traffic congestion. The HTCCR protocol involves 4.1% less delay for the ‘$C_{1}$’ and ‘$C_{2}$’ RT packets as it forwards packets in a pre-defined time before NRT data.

From Figure 15, it is evident that the BS in the HTCCR protocol receives an increased number of ‘$C_{1}$’ and ‘$C_{2}$’ RT packets compared with ‘$C_{3}$’ and ‘$C_{4}$’ NRT packets. It offers an increased speed of packet delivery and is suitable for diverse classes of traffic. The average throughput of RT is 10.4% better when compared with NRT when the network is highly congested. Packets that do not reach the threshold involve a lower number of hops and hence less delay. The throughput of ‘$C_{1}$’ is the maximum, while ‘$C_{2}$’ involves the least delay compared with other traffic. Though ‘$C_{2}$’ involves less delay, its throughput is reduced as reliability is not considered.

## 8. Conclusions

In this paper, a novel distributed clustering method to determine the optimal CH for every cluster in WSNs is proposed so as to maintain the trade-off between average delay and energy. The nodes join clusters where CHs are selected based on costs related to trust, buffers, waiting time, traffic load, energy, and distance. Further, a novel cost function is offered for an inter-cluster multi-hop routing protocol depending on the proposed delay model. A multi-hop routing protocol is offered to support efficient transmission from CHs to the sink involving lower energy cost, which is linked to the limitations of end-to-end delay. Further, to reduce the delay and energy overheads, XOR encoding and decoding, which enhances the network lifetime and reduces energy consumption, delay, and overheads, is included. Priority-based data delivery is included in the proposed scheme to avoid network congestion. The performance of the proposed protocol is compared with that of the existing TBSEER, CTRF, and TAGA protocols in terms of PDR, throughput, average delay, overhead, detection ratio of malicious nodes, packet loss, and energy. It is seen that the proposed HTCCR protocol outperforms the benchmarked protocols considered in this paper. The proposed HTCCR protocol involves 2.5, 2.3, and 1.7 times less delay; an 18.1%, 12.5%, and 5.5% better detection ratio; 2.9, 2.6, and 1.8 times less energy; a 2.2, 1.9, and 1.5 times lesser PLR; a 14.5%, 10.5%, and 5.2% better PDR; a 30.7%, 28.5%, and 18.4% better throughput; and 2.27, 1.91, and 1.66 times less routing overheads in comparison to the TBSEER, CTRF, and TAGA protocols, respectively. The HTCCR protocol involves 4.1% less delay for the ‘$C_{1}$’ and ‘$C_{2}$’ RT packets, and the average throughput of RT is 10.4% better when compared with NRT.

## Figures and Tables

**Figure 1 sensors-25-00864-f001:**
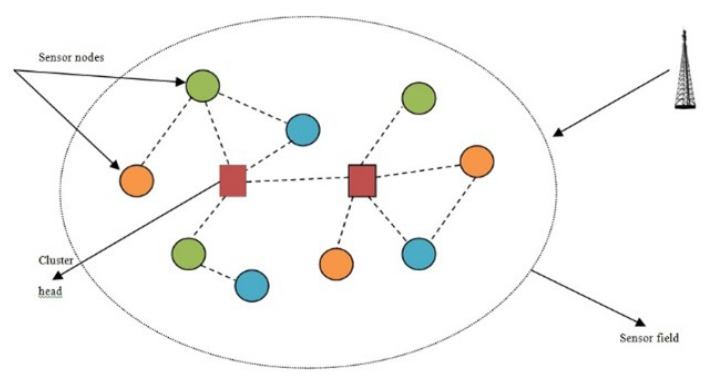
Data communication in WSN.

**Figure 2 sensors-25-00864-f002:**
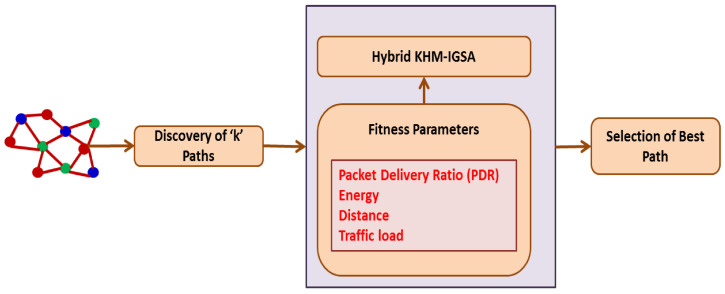
Overall flow of proposed work.

**Figure 3 sensors-25-00864-f003:**
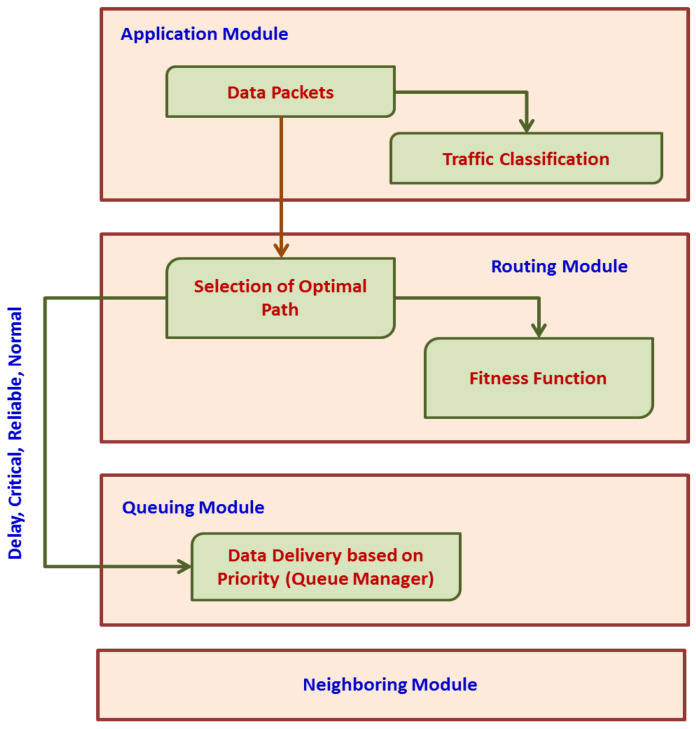
Traffic categorization module.

**Figure 4 sensors-25-00864-f004:**
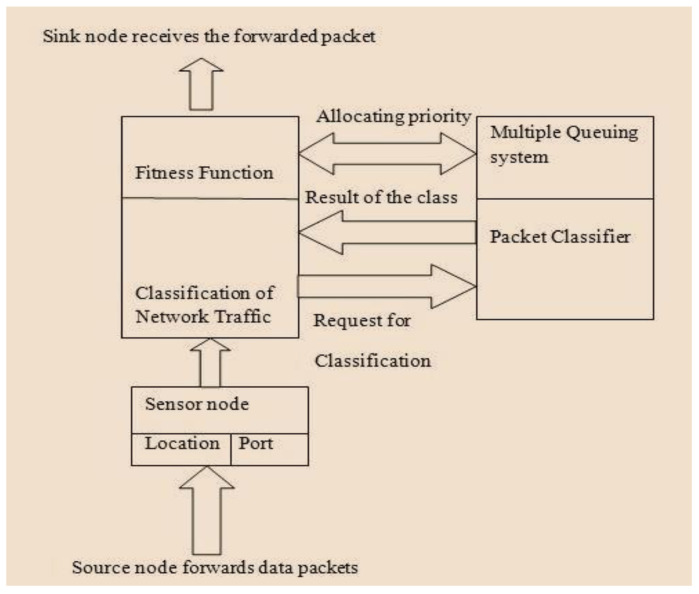
Network traffic classification model.

**Figure 5 sensors-25-00864-f005:**
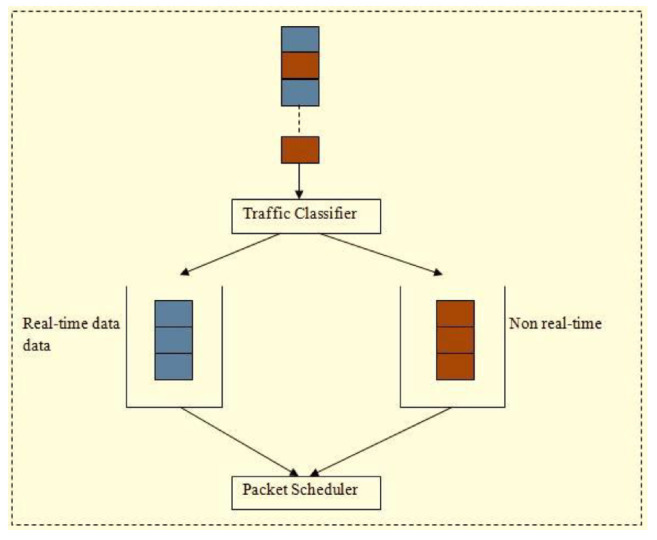
Traffic categorization model.

**Figure 6 sensors-25-00864-f006:**
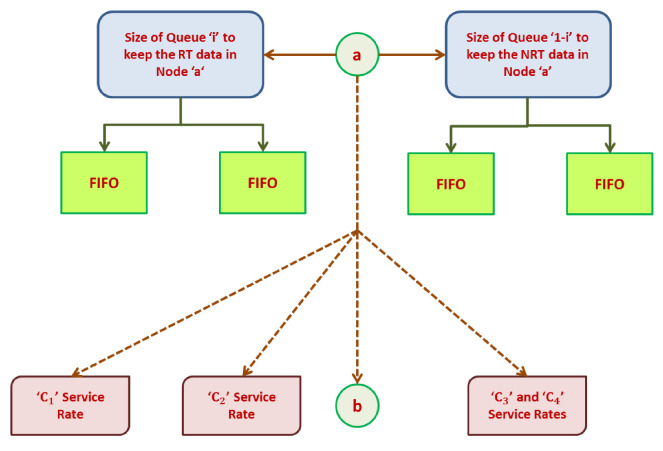
Service rate and size of queue in a node.

**Figure 7 sensors-25-00864-f007:**
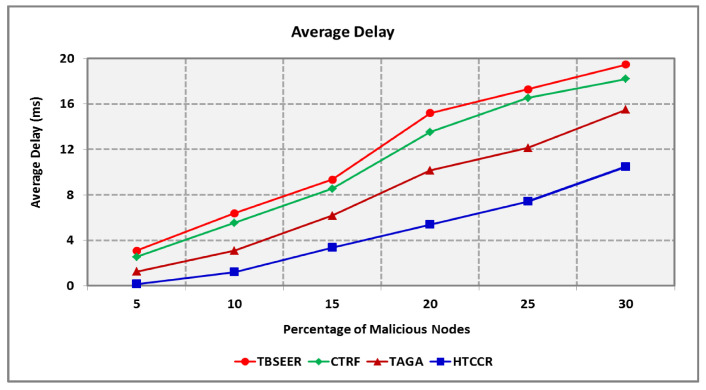
Average delay for a varying percentage of malicious nodes.

**Figure 8 sensors-25-00864-f008:**
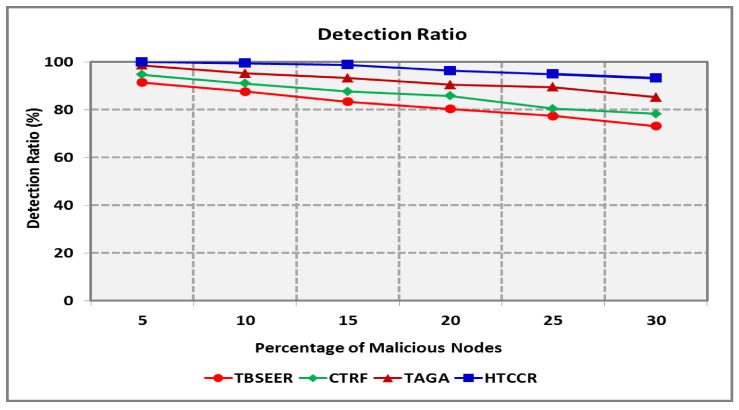
Detection Ratio for Varying Percentage of Malicious Nodes.

**Figure 9 sensors-25-00864-f009:**
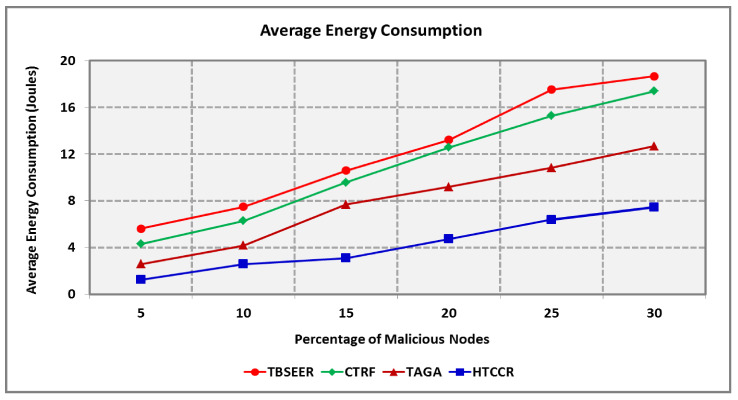
Energy Consumption for Varying Percentage of Malicious Nodes.

**Figure 10 sensors-25-00864-f010:**
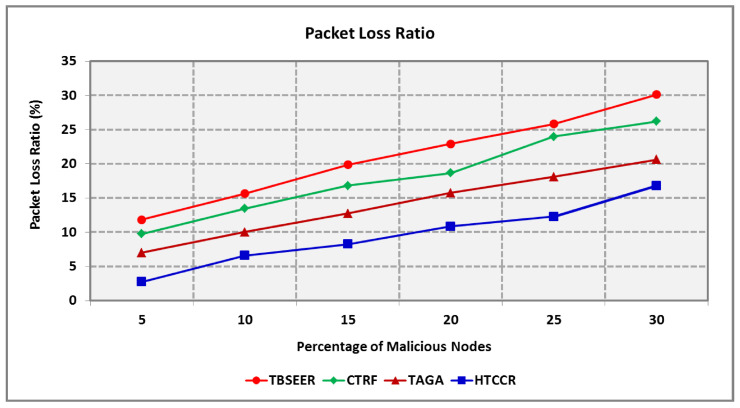
Packet Loss Ratio for Varying Percentage of Malicious Nodes.

**Figure 11 sensors-25-00864-f011:**
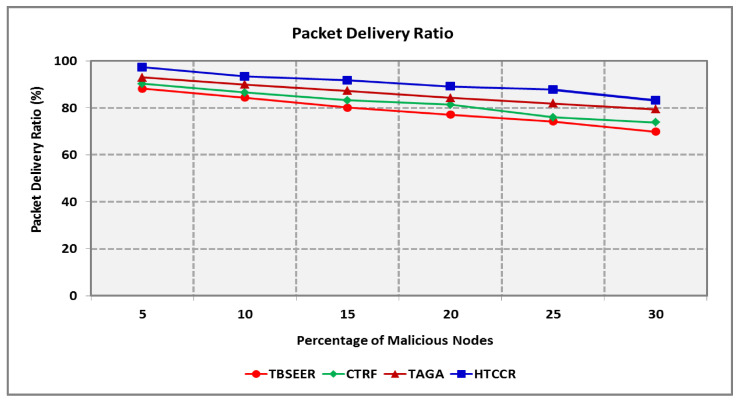
Packet Delivery Ratio for Varying Percentage of Malicious Nodes.

**Figure 12 sensors-25-00864-f012:**
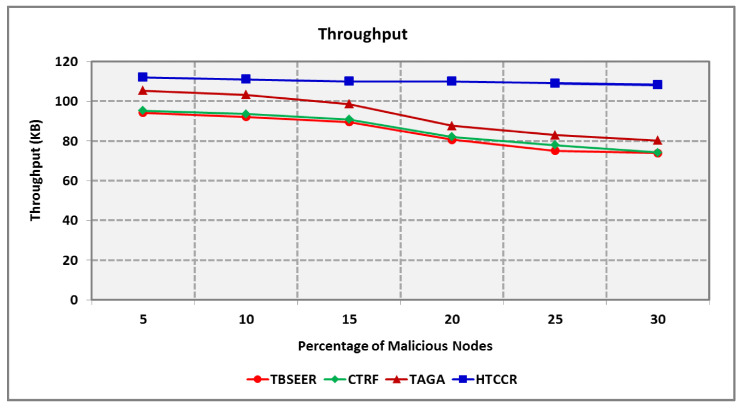
Throughput for Varying Percentage of Malicious Nodes.

**Figure 13 sensors-25-00864-f013:**
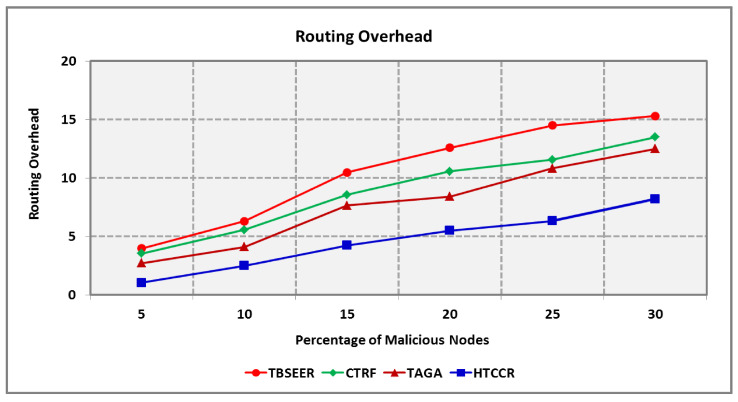
Routing Overhead for Varying Percentage of Malicious Nodes.

**Figure 14 sensors-25-00864-f014:**
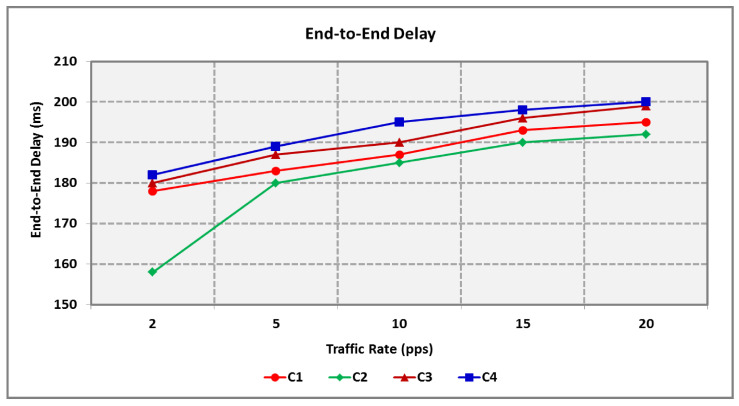
End-to-End Delay for Varying Traffic Rates.

**Figure 15 sensors-25-00864-f015:**
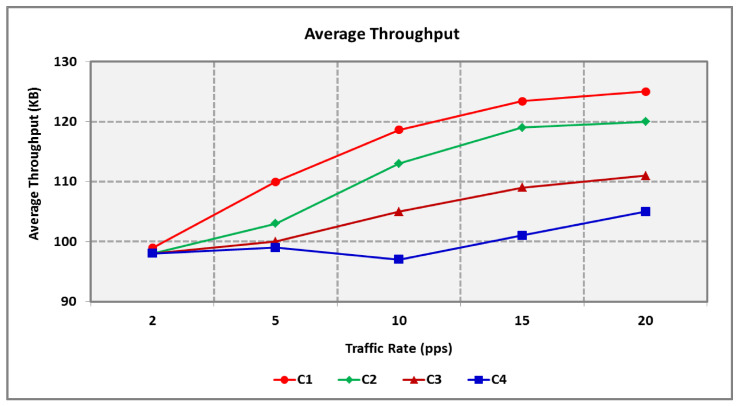
Throughput for Varying Traffic Rates.

**Table 1 sensors-25-00864-t001:** Definition of notations.

Notation	Definition
RSSI	Received signal strength indicator
$Dist_{s,i}$	Signal system strength between source node S and current node i
$G_{i}$	$i^{th}$ node in the graph
$Trust_{DT}\left(i\right)$	The degree of direct trust which is present inside a node at time *i*
$Trust_{IDT}\left(i\right)$	The degree of direct trust which is present inside adjacent nodes at time *i*
ω,γ	Fitness factors
$N_{d}$	Number of neighbouring nodes
τ	Fitness factor which is utilized to stability of the measurement in the prevailing and past iterations that are stated between the range 0<τ<1
$N_{Ce}$	Energy consumption of sensor node
$N_{TXe}$ and $N_{RXe}.$	Energy Spent during transmission and Reception
$N_{LXe}$ and $N_{SXe}.$	Energy Spent during idle and sleep mode
$N_{Re}$	Residual energy present in a node
$N_{le}$	Initial energy of a node
$T_{i}\left(k\right)$	Buffer length of *k*th node
congTh(min)	Minimum threshold for congestion
congTh(Max)	Maximum threshold for congestion
$dist_{1}$	The distance between current source node and potential next node
$dist_{2}$	The distance between potential next node to sink
α,β	Adjustment parameters
$T_{wmax}$	Pre-defined maximum waiting time
$Avg(S_{i}^{k})$ and $S_{n}$	Average speed of the nearest nodes inside transmission range and the speed of the node
δ, ε, ψ, γ, υ and τ	Adjustment parameters lie between 0 to 1

**Table 2 sensors-25-00864-t002:** Type of data traffic in WSNs.

Parameters to Be Observed	Type of the Packet Traffic
Scalar data (Pressure, Heat)	RT traffic
Scalar data (Humidity)	NRT traffic
Periodic monitoring of videos	NRT traffic
Tracking of image/videos	RT traffic

**Table 3 sensors-25-00864-t003:** Importance and weight levels.

Fitness Factor	$C_{1}$ $C_{2}$	$C_{3}$	$C_{4}$	
$N_{RE}^{i}$	High	Average	High	Same
$Trust_{xy}^{*}$	High	Average	High	Same
$T_{dist}$	High	High	Low	Same
$T_{s}^{k}$	Average	Low	Average	Same

**Table 4 sensors-25-00864-t004:** Simulation parameters.

Parameter	Value
Simulator	NS-3.25
Topology	Random node positioning with a mobile sink
Number of nodes	500
Size of the packet	3000 bits
Size of the control packet	300 bits
Initial energy	0.5 joules
Eelec, Etx, Erx	50 nJ/bit
Rounds	1200
Trust threshold	0.5
Protocols considered	HTCCR (proposed), TBSEER, CTRF, and TAGA (Existing)

## Data Availability

Data can be made available upon reasonable request by contacting email/contact information.

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
