# Peer review of "Hybrid Reliable Clustering Algorithm with Heterogeneous Traffic Routing for Wireless Sensor Networks"

_sensors, 2025, doi:10.3390/s25030864_

Round 1
Reviewer 1 Report
Comments and Suggestions for Authors
The paper introduces a Trust-based Congestion-aware Cluster Routing (HTCBR) algorithm, aiming to detect attacker nodes, reduce congestion, and optimize routing through clustering in WSNs. Although details of the proposed solution are presented, the paper lacks clarity and conciseness. Comprehensive revision is required to improve coherence, readability, logical flow, citations and referencing, and eliminate redundancy. Furthermore, a stronger link between the objectives in the abstract/introduction and the results in the evaluation is needed to support the findings more robustly. Enhancements should focus on improving clarity, avoiding ambiguity, and reinforcing relevance. Below are my comments:
- The word "Intelligent" in the title appears irrelevant to the content of the paper. Please refine accordingly.
- Adjust the abbreviation to “Hybrid Trust-based Congestion Aware Cluster Routing (HTCBR).” In the abstract.
- Include key numerical results in the abstract along with specifying the evaluation metrics.
- In Section 3, expand the discussion to explain how the solution ensures "congestion-free trust-aware secure routing," and reflect this in the evaluation based on obtained results.
- You did not provide references for the algorithms used in the comparative analysis. Please include references, and you are supposed to compare HTCBR with existing routing solutions addressing similar issues, specifically enhanced security, as claimed.
- The majority of references appear outdated. Please update the introduction and literature review with more recent references from the past five years to reflect the rapid advancements in WSNs.
- Discuss how your solution relates to the following recent energy-conservation approaches in WSNs concerning the problem you addressed:
· DOI: 10.1109/ICSC60394.2023.10441568
· DOI:10.3390/s24175630
· DOI:10.1109/ASET60340.2024.10708695
· DOI:10.1002/ett.5039
- Use a consistent referencing style to ensure alignment between in-text citations and the reference list (e.g., IEEE).
- The paper contains language and grammar errors. Please revise for grammatical accuracy and tense consistency (e.g., “The computed fitness function will be applied”).
- Improve clarity and resolution of Figures 2, 3, 6, 7–14 to ensure label readability.
- Rewrite the algorithm on pages 19-20 using proper design tools, and assign it a number (e.g., Algorithm 1).
- Present Tables 2, 3, and 4 in table format rather than images, similar to Table 1, to avoid plagiarism concerns.
- You mentioned algorithms like LEACH, TRPM, and LEACH-FT in the text without providing citations. Please cite each solution mentioned throughout the paper, including the source and publication date, to improve clarity.
- Please clarify how the values in Table 4 were determined—whether they are arbitrary or based on prior studies. If they are based on previous research, provide appropriate citations.
- Your simulation uses 100 nodes. How can you test the scalability of your solution? Please discuss its applicability to real-world scenarios involving thousands of nodes in WSNs.
- Align figures with references in text (e.g., “the graph in Figure 7” instead of “the above figure”).
- In the evaluation structure, please add a "Results and Discussion" subsection under "Performance Evaluation" to enhance clarity and organization.
- Please elaborate on the claim that “delay is directly proportional to malicious nodes” in the context of the LEACH protocol. Is this accurate for LEACH? If so, please cite relevant literature to support this claim.
- The analysis of results requires further improvement. Please strengthen the discussion on how the results align with the research objectives stated in the abstract and introduction.
- In Conclusion, reflect key numerical results to reinforce the paper’s findings.
I hope this feedback will help improve your manuscript so that it meets publication standards and requirements.
Comments on the Quality of English LanguageThe paper needs improvement in English writing. Please enhance clarity, coherence, and overall language quality
Reviewer 2 Report
Comments and Suggestions for Authors
The authors of this paper consider the problem of congestion control, utilization of trust for enhanced security and incorporation of clustering technique in the field of wireless sensor networks in which the detection of attackers is significant to reduce the congestion inside the network. To tackle this problem, the authors propose a Trust based Congestion Aware Cluster Routing algorithm (HTCBR) that effectively detects the attacker nodes and reduces the congestion with optimal routing using a clustering method. The authors claim that previous algorithms for congestion control in Wireless Sensor Networks failed to consider the attacker behavior and drawbacks in the network congestion. By comparing the novel HTCBR with the existing algorithms in terms of average delay, PDR, throughput, detection ratio, packet loss and energy through simulations, the authors prove that the presented HTCBR achieves better performance.
The present paper considers the congestion problem in Wireless Sensor Networks which initiates network congestion from the source to the sink in the upstream direction if the speed of transmission and data processing time falls behind the incoming traffic speed. This problem could lead to increased latency, packet dropping, buffer overflows, energy wastage, and reduced throughput of the network and decreased quality of service. Thus, detecting these congestions and controlling them constitute vital needs for WSNs. The authors of this paper contribute to the control of these congestions via energy efficient and dependable congestion detection, control and transport mechanisms to optimize QoS needs and network resources.
The major contributions of the authors in this paper are: to provide higher detection accuracy of malicious nodes to isolate them from the routing path of the data by utilizing trust concept. Next, a hybrid k-harmonic means and gravitational search algorithm to function as an effective cluster based and congestion-free, secure routing. Finally, they incorporate a priority based delivery to the presented secure routing model by monitoring some parameters in the traffic classification model. The type of traffic is identified through these parameters in real-time or non-real-time. Every class of traffic has been allocated with dissimilar weights. The queuing model carries a threshold value to keep the arriving data packets on the basis of priority queues. As such, network congestion is managed effectively to improve the overall performance of the network.
The steps followed by the authors are clear and the paper does not require any further improvements in terms of methodology. Yet, some amendments need to be done on the paper content regarding the language of the paper in some parts that needs improvement in terms of language. A complete proofreading should be done.
The conclusions need to be rewritten to well summarize the work results and highlight the key findings.
The list of references is adequate and completely summarizes the state of art findings on the topic of the paper. Yet, it could be enhanced by adding more and recent items.
Comments on the Quality of English LanguageThe language of the paper needs improvement. A complete proofreading is needed.
